# Severe mental illness diagnosis in English general hospitals 2006-2017: A registry linkage study

Hassan Mansour[1]*, Christoph Mueller[2,3], Katrina A. S. Davis[2,3], Alexandra Burton[1], Hitesh Shetty[3], Matthew Hotopf[2,3], David Osborn[1,4], Robert Stewart[2,3], Andrew Sommerlad[1,4]

1 Division of Psychiatry, University College London, United Kingdom, 2 King's College London, Institute of Psychiatry, Psychology and Neuroscience, London, United Kingdom, 3 South London and Maudsley NHS Foundation Trust, London, United Kingdom, 4 Camden and Islington NHS Foundation Trust, London, United Kingdom

* hassan.mansour.17@ucl.ac.uk

**Data Availability Statement:** All relevant aggregate data are found within the paper. The data used in this work have been obtained from the Clinical Record Interactive Search (CRIS), a system that

## Abstract

### Background

The higher mortality rates in people with severe mental illness (SMI) may be partly due to inadequate integration of physical and mental healthcare. Accurate recording of SMI during hospital admissions has the potential to facilitate integrated care including tailoring of treatment to account for comorbidities. We therefore aimed to investigate the sensitivity of SMI recording within general hospitals, changes in diagnostic accuracy over time, and factors associated with accurate recording.

### Methods and findings

We undertook a cohort study of 13,786 adults with SMI diagnosed during 2006–2017, using data from a large secondary mental healthcare database as reference standard, linked to English national records for 45,706 emergency hospital admissions. We examined general hospital record sensitivity across patients' subsequent hospital records, for each subsequent emergency admission, and at different levels of diagnostic precision. We analyzed time trends during the study period and used logistic regression to examine sociodemographic and clinical factors associated with psychiatric recording accuracy, with multiple imputation for missing data.

Sensitivity for recording of SMI as any mental health diagnosis was 76.7% (95% CI 76.0–77.4). Category-level sensitivity (e.g., proportion of individuals with schizophrenia spectrum disorders (F20-29) who received any F20-29 diagnosis in hospital records) was 56.4% (95% CI 55.4–57.4) for schizophrenia spectrum disorder and 49.7% (95% CI 48.1–51.3) for bipolar affective disorder. Sensitivity for SMI recording in emergency admissions increased from 47.8% (95% CI 43.1–52.5) in 2006 to 75.4% (95% CI 68.3–81.4) in 2017 (ptrend < 0.001). Minority ethnicity, being married, and having better mental and physical health were associated with less accurate diagnostic recording. The main limitation of our study is the potential for misclassification of diagnosis in the reference-standard mental healthcare data.

has been developed for use within the NIHR Mental Health Biomedical Research Centre (BRC) at the South London and Maudsley NHS Foundation Trust (SLaM). It provides authorized researchers with regulated access to anonymized information extracted from SLaM's electronic clinical records system. Individual-level data are restricted in accordance to the strict patient led governance established at South London and The Maudsley NHS Foundation Trust, and by NHS Digital for the case of linked data. Data are available for researchers who meet the criteria for access to this restricted data: (1) SLaM employees or (2) those having an honorary contract or letter of access from the trust. For further details, and to obtain an honorary research contract or letter of access, contact the CRIS Administrator at cris.administrator@kcl.ac.uk.

**Funding:** The authors received no specific funding for this work.

**Competing interests:** I have read the journal's policy and the authors of this manuscript have the following competing interests: The data resource is funded by the National Institute for Health Research (NIHR) Biomedical Research Centre at South London and Maudsley NHS Foundation Trust and King's College London. HM, AB, DO, and AS are supported by the UCLH NIHR Biomedical Research Centre. KD is funded by the NIHR Biomedical Research Centre at South London and Maudsley NHS Foundation Trust and King's College London. DO is in part supported by the NIHR Applied Health Research Collaboration (ARC) North Thames at Bart's Health NHS Trust. RS declares receipt of research support within the last 5 years from Roche, Janssen, GSK, and Takeda. AS was funded by a fellowship from the Wellcome trust during the submitted work.

**Abbreviations:** ADL, activity of daily living; EHR, electronic health record; HES, Hospital Episode Statistics; HoNOS, Health of the Nation Outcome Scale; ICD-10, International Statistical Classification of Diseases and Related Health Problems, Tenth Revision; IQR, interquartile range; OR, odds ratio; SLaM, South London and Maudsley National Health Service Foundation Trust; SMI, severe mental illness.

## Conclusions

Our findings suggest that there have been improvements in recording of SMI diagnoses, but concerning under-recording, especially in minority ethnic groups, persists. Training in culturally sensitive diagnosis, expansion of liaison psychiatry input in general hospitals, and improved data sharing between physical and mental health services may be required to reduce inequalities in diagnostic practice.

## Author summary

### Why was this study done?

- People with severe mental illness (SMI) have increased mortality and morbidity, largely due to preventable medical conditions, and these disparities have the potential to be ameliorated through better healthcare integration.
- Accurate recognition of SMI during hospital admissions can be critical as it allows continuity of previous pharmacological and supportive treatments and tailoring of inpatient and discharge care to individual needs.

### What did the researchers do and find?

- We examined the hospital discharge records of 13,786 individuals with SMI diagnosis from a mental health service, who had 45,706 admissions to English general hospitals between 2006 and 2017.
- We found that a psychiatric condition is recorded in around two-thirds of general hospital admissions of people with SMI. Recording of SMI diagnosis increased between 2006 and 2017.
- However, people from ethnic minority and married backgrounds were less likely to have psychiatric diagnosis recorded. Similarly, those with less severe mental or physical health symptoms were also less likely to have diagnosis recorded.

### What do these findings mean?

- Despite improvements over the past decade, inequities related to ethnicity remain. Policy-makers and clinicians should endeavor to improve recognition and recording of SMI in general hospital settings to promote integrated physical and mental healthcare.
- A limitation of our study is that our use of electronic health records for the reference-standard means that some people with SMI may have been misclassified.

## Introduction

Severe mental illnesses (SMI), defined as schizophrenia spectrum and bipolar affective disorders, have lifetime prevalence of around 0.5% and 1% respectively [1, 2], and are associated with several physical comorbidities that increase the risk of general hospital admission [3, 4]. Life expectancy is 10–15 years lower for people with SMI [5], and this health gap is increasing over time [6], with SMI contributing to 3.5% of worldwide years lost to disability [7]. The 2019 Lancet Commission on Physical Health of People with Mental Illness identified the need for multidisciplinary approaches to multimorbidity in people with mental illness [8]. An important aspect of such healthcare integration is detection of psychiatric conditions in healthcare settings to allow continuity of previous care such as medication and tailoring of inpatient and discharge plans to account for comorbidity [9]. However, the separation of physical and mental healthcare in the United Kingdom, with different hospital settings using discrete electronic health records (EHRs), may inhibit this.

There has been limited research into recognition of mental health conditions within secondary physical healthcare settings. One study investigated SMI recording in primary care [10], and another investigated recognition specifically of deliberate self-poisoning within one UK general hospital's EHRs [11]; both were conducted over 20 years ago. A further study examined recording of alcohol disorders in primary and secondary care [12]. Other studies have investigated accuracy of psychiatric, rather than general hospital, records, in which recognition of mental illness is likely to be higher [13]. No studies have investigated time-trend changes or factors associated with systematic differences in SMI recording. Understanding such changes may inform future approaches to improving illness recording and elucidate potential biases in hospital records, which are being increasingly used for case-ascertainment in epidemiological studies [14].

We therefore aimed to evaluate the accuracy of SMI recording during English general hospital admissions of a person with preexisting SMI who is admitted to a general hospital for any health condition, using a large secondary mental healthcare database to identify people with SMI, and linked national general hospital data to assess diagnostic recording. Our specific objectives were to calculate the sensitivity of SMI diagnosis in general hospital records, evaluate time-trend changes of SMI diagnosis recording in general hospital records from 2006 to 2017, and examine association of sociodemographic and clinical factors with accuracy of psychiatric recording in general hospital records.

## Methods

### Study design and participants

We undertook a cohort study, using data from the South London and Maudsley National Health Service (NHS) Foundation Trust (SLaM), one of Europe's largest secondary mental healthcare trusts providing support to around 1.36 million people living in 4 ethnically diverse communities in South London, UK (Croydon, Lambeth, Lewisham, and Southwark). We used the "Clinical Record Interactive Search" (CRIS) data extraction tool, which enables construction of databases suitable for research by identifying, retrieving, and linking a pseudonymized version of SLaM patient records for over 450,000 individuals [15]. CRIS uses natural language processing (NLP) algorithms developed on General Architecture for Text Engineering (GATE) software [16] to extract information from unstructured fields of the clinical record. SLaM data were linked using deterministic matching procedures to NHS Digital's Hospital Episode Statistics data source to identify admissions to any English hospital. Oxfordshire Research Ethics Committee C (18/SC/0372) approved these resources for secondary analysis.

The terms of the ethical approval do not require consent to be provided, but all participants have the right to opt out of data use at any time. This study is reported as per the Strengthening the Reporting of Observational Studies in Epidemiology (STROBE) guideline (S1 STROBE Checklist) [17].

These data were selected to generate "reference-standard" SMI diagnoses because they included information from the predominant diagnostic and treatment service for mental illnesses in the catchment area. Patients were diagnosed after assessment by mental health staff, e.g., a psychiatrist, nurse, or psychologist, according to the International Statistical Classification of Diseases and Related Health Problems, Tenth Revision (ICD-10) [18], which is the predominant diagnostic framework in the UK.

We included study participants who (1) had clinical contact with SLaM mental health services between 1 January 2006 and 31 March 2017 while aged 18 or over, (2) were diagnosed at any time with schizophrenia (ICD-10 code F20), schizotypal disorder (F21), delusional disorder (F22), schizoaffective disorder (F25), other nonorganic psychotic disorder (F28), unspecified nonorganic psychosis (F29), manic episode (F30), or bipolar affective disorder (F31), including those described in clinical coding or records as "probable" or "in remission," and (3) were admitted to an English NHS general (nonpsychiatric) hospital after the first diagnosis of SMI in CRIS. Participants were identified from diagnoses in structured EHR diagnostic fields or unstructured text. Participants were assigned to only 1 diagnostic category following the "hierarchy" of psychiatric diagnoses [19] (e.g., an individual diagnosed sequentially with schizophrenia [F20] and a manic episode [F30] would be assigned to F20 group), as in previous studies [20]. We did this to avoid double-counting patients who may have been diagnosed at different times within the reference-standard database as having a schizophrenia-like disorder and bipolar disorder; we judged that this most likely represented evolving clinical opinion rather than coexistence of the 2 disorders, which is clinically unlikely [21].

## Outcomes

We obtained outcome data from Hospital Episode Statistics (HES) records, which include clinical diagnoses according to ICD-10 criteria from admissions to any hospital within England [22]. The HES database is primarily designed to allocate payment to hospitals for the care they provide, and its secondary uses are for research and health service planning. The index date for our study was the first SMI diagnosis within SLaM, and we obtained data from HES on all subsequent admissions of included patients to general (i.e., nonpsychiatric) hospitals, including dates of admission and discharge, admission method (emergency, i.e., unplanned admission, or elective, such as admission for renal dialysis, wound dressing, chemotherapy, or elective surgery), and up to 20 primary and secondary recorded diagnoses. Our outcome of interest was recording of psychiatric illness in any of the 20 recorded primary or secondary diagnoses.

Diagnoses recorded in HES include those clinically identified by hospital staff during admissions, those derived from preexisting clinical records from secondary mental health trusts or previous hospital medical records, or those obtained following communication with primary care. Separate EHRs are used in different hospital trusts, meaning that diagnoses are not automatically entered in general hospital records from secondary mental healthcare or primary care records. Some EHRs may automatically populate diagnosis fields with previously recorded chronic conditions, although there are no data available to determine the extent of this practice.

## Covariates

We obtained age, sex, ethnicity (White, Mixed, Asian or British Asian, Black or Black British, and other) and marital status (single, married or cohabiting, divorced or separated, and

widowed) from SLaM records data. We used the Index of Multiple Deprivation (IMD) [23] to rate neighborhood-level socioeconomic deprivation. Clinical presentation was derived using the Health of the Nation Outcome Scale (HoNOS), which is a clinician-rated measure routinely applied in UK mental health services to assessed patients with good validity and adequate reliability [24]. HoNOS comprises of 12 subscales, each rated on a 5-point Likert scale with higher values indicating more severe problems; we dichotomized these into 0–1, indicating no or minor problems, and 2–4, indicating more severe problems. As we aimed to assess association of mental illness severity with diagnostic recording, we combined the subscales reflecting mental health symptoms (agitation, self-injury, drug/alcohol use, cognitive impairment, delusions/hallucinations and depressed mood) into an ordinal scale indicating 0 symptoms, 1 current mental health symptom, 2 current mental health symptoms, and 3+ current mental health symptoms, as in previous studies [25]. We also used the physical illness and activity of daily living (ADL) impairment subscales as covariates. All covariates were derived from time closest to first general hospital admission.

## Analysis

Our prospective analysis plan is in S1 Text. We described sociodemographic characteristics according to whether SMI had ever been recorded in hospital records using chi-squared tests for categorical data and independent *t* tests for continuous data. We described the primary diagnoses for admissions.

1. **Sensitivity of general hospital SMI diagnosis.** We examined sensitivity of psychiatric diagnosis recording at patient-level (proportion of people with SLaM SMI diagnosis who ever had a mental illness recorded in their complete general hospital records as a primary or secondary diagnosis) and emergency admission-level (proportion who have a mental illness recorded in primary or secondary diagnoses during each emergency admission). We chose to examine emergency admissions as nonemergency admissions are usually recurrent brief admissions, which we considered were unlikely to warrant a full diagnostic assessment [26]. In response to peer-reviewer comments, we additionally calculated (1) patient-level sensitivity according to the number of previous hospital admissions (0, 1–5, ≥6) and (2) admission-level sensitivity according to the primary diagnosis recorded for each hospital admission (grouped into ICD-10 categories). We report our sensitivity calculations at different levels of diagnostic accuracy. Our primary analyses calculated sensitivity at the level of any psychiatric diagnosis (proportion of individuals with SLaM diagnosis of any F20-31 disorder who receive any psychiatric diagnosis [F00-99] in HES). We also examined category-specific sensitivity (e.g., proportion of individuals with schizophrenia spectrum disorders [F20-29] who received any F20-29 diagnosis in HES) and disorder-specific diagnosis (e.g. proportion of individuals with schizophrenia diagnosis [F20] who specifically received an F20 diagnosis in HES). We calculated 95% confidence intervals for sensitivity using the Clopper–Pearson method [27].

2. **Time-trend changes from 2006 to 2017.** We calculated sensitivity for each participant's first emergency hospital admission following SMI diagnosis stratified by year of admission and used Cochran–Armitage test to examine sensitivity changes over time [28].

3. **Association of sociodemographic and clinical factors with psychiatric diagnosis being unrecorded.** We calculated the association of sociodemographic and clinical characteristics with unrecorded diagnosis (no psychiatric diagnosis ever being recorded at a patient level) using multivariable logistic regression. Included variables were age, sex, ethnicity, marital status, IMD, number of mental illness symptoms, physical illness, ADL impairment, and

number of hospital admissions (log-transformed because of the skewed distribution). To explore the impact of missing data, we conducted 2 non-prespecified sensitivity analyses. We first repeated the analysis without HoNOS variables because these data were missing for 21% of the cohort. We also used multiple imputation with chained equations to create 20 imputed datasets using STATA's mi package by replacing missing values using a model constructed from all available covariates and outcome variables; our imputation used predictive mean matching for continuous data and logistic regression for categorical data. We then conducted logistic regression on each imputed dataset before combining coefficients using Rubin's rules [29]. The fraction of missing information after imputation was 0.33, so 20 imputed datasets is likely to give replicable estimates of standard error [30].

All analyses were undertaken using STATA SE version 15 (Stata Corp. https://www.stata.com/).

## Results

We identified 28,832 individuals with F20-31 diagnosis who were seen by SLaM between 1 January 2006 and 31 March 2017 (Fig 1). We excluded 2,410 because of diagnosis of acute and transient psychotic disorder (F23) or induced delusional disorders (F24). A further 12,636 individuals were excluded as they had no hospital admissions in the study period. Our final sample included 13,786 individuals who had 45,706 emergency admissions (Fig 1). Primary ICD-10 diagnoses are described in S1 Table; mental disorder was the primary diagnosis in 7.9% of emergency hospital admissions.

The mean age was 46.9 years, 51% of participants were female, and the majority were single (64.7%) (Table 1). Most were White ethnicity (59.5%), and the largest ethnic minority group was Black African/Caribbean (27.7%). Three-quarters (10,574, 76.7%) of participants had a schizophrenia spectrum disorder, and 3,212 (23.3%) had bipolar disorder. At the time closest to first general hospital admission, many participants displayed no (24.8%) or one (25.1%) significant mental health symptoms; full information on prevalence of clinical problems in individual HoNOS domains is in S2 Table. The median number of hospital admissions was 3 admissions (interquartile range [IQR] 1, 5) with median duration of follow-up being 7.19 years [IQR 0.4, 10]; 3,047 people died during follow-up.

### Sensitivity of general hospital SMI diagnosis

We found that 10,574 of 13,786 people with SMI had any psychiatric illness recorded during their subsequent general hospital admissions, meaning that sensitivity at the level of each individual patient's complete hospital records was 76.7% (95% CI 76.0–77.4) (Table 2). Patient-level sensitivity was 57.3% (55.8–58.8) for patients with only 1 admission, 80.7% (79.7–81.6) for those with 2–5 admissions, 92.8% (91.5, 93.9) for those with 6–10 admissions, and 96.7 (95.7, 97.6) for those with 11+ admissions.

Sensitivity is lower when examining more specific diagnosis. Sensitivity was 56.4% (55.4–57.4) for category-specific ICD-10 diagnosis for schizophrenia spectrum disorder (F20-29) and 49.7% (48.1–51.3) for bipolar affective disorders (F30-31). Full results for disorder-specific recording can be found in S3 Table. Patient-level sensitivity for schizophrenia (F20) was 45.1% (44.5–45.7) and 40.4% (39.5–41.3) for bipolar affective disorder (F31). Disorder-specific sensitivity for other rarer conditions such as schizoaffective or schizotypal disorder was lower.

For admission-level recordings, 32,033 of 45,706 emergency hospital admissions had any psychiatric diagnosis recorded meaning that sensitivity was 70.1% (69.7–70.5) (Table 3). Recording of any psychiatric diagnosis was 71.1% (70.7–71.6) for schizophrenia spectrum

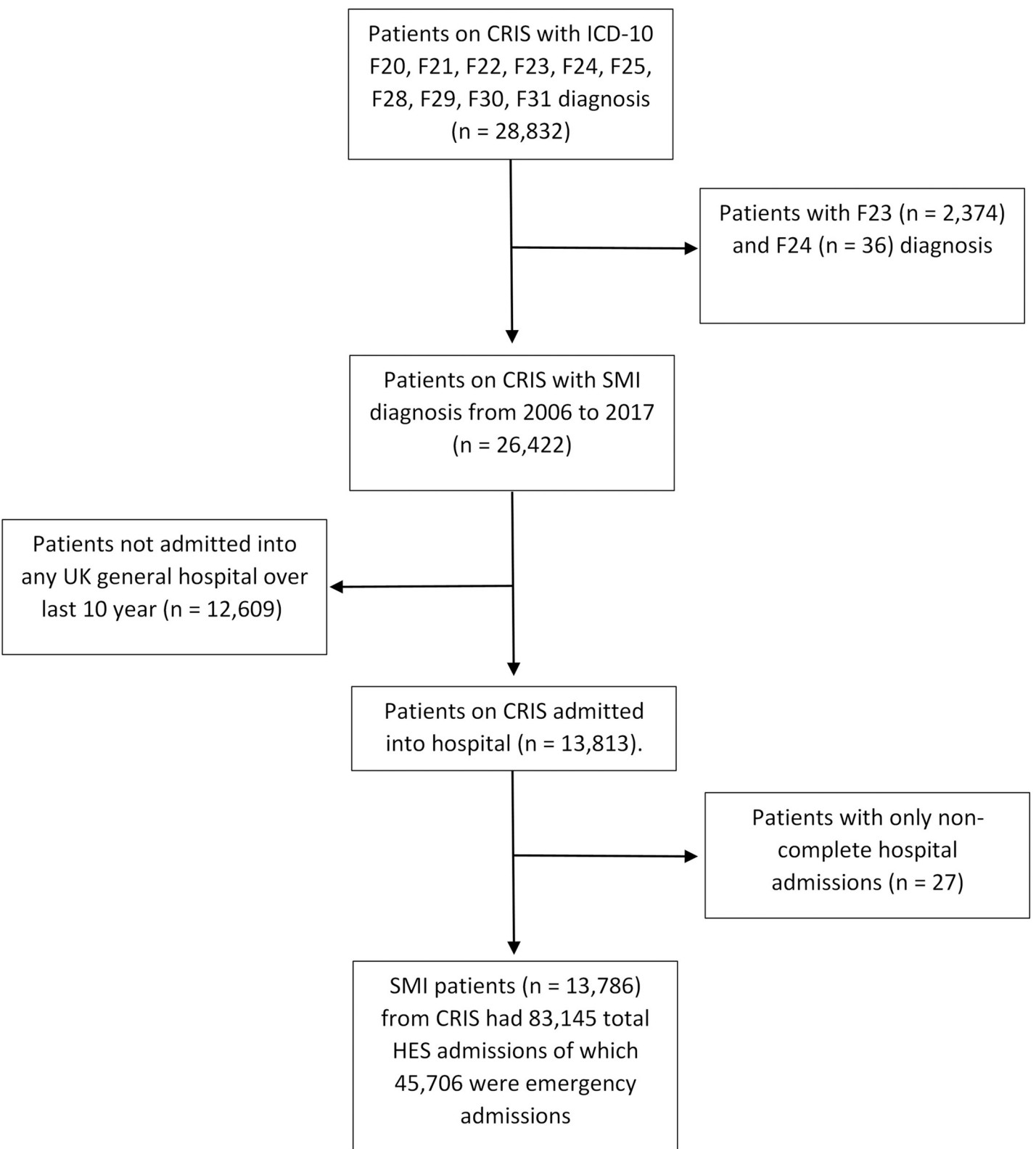

**Fig 1. Cohort of individuals with SMI who were also admitted into general hospitals.** CRIS, Clinical Record Interactive Search; F20, schizophrenia; F21, schizotypal disorder; F22, delusional disorder; F23, acute and transient psychotic disorder; F24, induced delusional disorders; F25, schizoaffective disorder; F28, other nonorganic psychotic disorder; F29, unspecified nonorganic psychosis; F30, manic episode; F31, bipolar affective disorder; HES, Hospital Episode Statistics; ICD-10, International Statistical Classification of Diseases and Related Health Problems, Tenth Revision; SMI, severe mental illness.

**Table 1. Sociodemographic and clinical characteristics of all participants, according to whether psychiatric diagnosis ever made in subsequent general hospital records.**

| Sociodemographic and Clinical Characteristics | | All Patients (n = 13,786) | | Psychiatric Diagnosis Recorded (n = 10,574) | | No Psychiatric Diagnosis Recorded (n = 3,212) | | Significance Test |
|---|---|---|---|---|---|---|---|---|
| | | N | % | N | % | N | % | |
| **Age**[a] | Mean (SD) | 46.9 (17.3) | | 47.7 (17.5) | | 44.4 (16.7) | | $t = 9.5, p < 0.001$ |
| | 18–24 | 1,279 | 9.3 | 933 | 8.8 | 346 | 10.8 | $\chi2 = 92.5, p < 0.001$ |
| | 25–34 | 2,669 | 19.4 | 1,957 | 18.5 | 712 | 22.2 | |
| | 35–44 | 3,108 | 22.5 | 2,310 | 21.9 | 798 | 24.8 | |
| | 45–54 | 2,368 | 17.2 | 1,823 | 17.2 | 545 | 17.0 | |
| | 55–64 | 1,968 | 14.3 | 1,586 | 15.0 | 382 | 11.9 | |
| | 65+ | 2,394 | 17.4 | 1,965 | 18.6 | 429 | 13.4 | |
| | *Missing* | 0 | | 0 | | 0 | | |
| **Sex** | Female | 7,036 | 51.0 | 5,352 | 50.6 | 1,684 | 52.4 | $\chi2 = 3.2, p = 0.072$ |
| | *Missing* | 0 | | 0 | | 0 | | |
| **Ethnicity** | White | 7,896 | 59.5 | 6,308 | 61.8 | 1,588 | 51.9 | $\chi2 = 99.2, p < 0.001$ |
| | Mixed | 326 | 2.5 | 244 | 2.4 | 82 | 2.7 | |
| | Asian | 677 | 5.1 | 505 | 4.9 | 172 | 5.6 | |
| | Black African/ Caribbean | 3,673 | 27.7 | 2,659 | 26.0 | 1,014 | 33.1 | |
| | Other | 703 | 5.3 | 498 | 4.9 | 205 | 6.7 | |
| | *Missing* | 511 | | 360 | | 151 | | |
| **Marital Status**[b] | Single | 8,217 | 64.7 | 6,362 | 64.9 | 1,855 | 64.1 | $\chi2 = 38.6, p < 0.001$ |
| | Married | 2,076 | 16.4 | 1,512 | 15.4 | 564 | 19.5 | |
| | Divorced | 1,621 | 12.8 | 1,281 | 13.1 | 340 | 11.7 | |
| | Widowed | 780 | 6.1 | 644 | 6.6 | 136 | 4.7 | |
| | *Missing* | 1,092 | | 775 | | 317 | | |
| **Mean deprivation score (SD)**[b] | | 30.0 (1.1) | | 30.0 (1.1) | | 30.1 (1.0) | | $t = -0.7, p = 0.467$[b] |
| | *Missing* | 467 | | 389 | | 78 | | |
| **Mental Health Symptoms (from HoNOS subscales)**[c] | No Symptoms | 3,427 | 31.2 | 2,465 | 29.4 | 962 | 37.2 | $\chi2 = 93.1, p < 0.001$ |
| | 1 Symptom | 3,458 | 31.5 | 2,611 | 31.1 | 847 | 32.8 | |
| | 2 Symptoms | 2,314 | 21.1 | 1,841 | 22.0 | 473 | 18.3 | |
| | 3+ Symptoms | 1,771 | 16.1 | 1,467 | 17.5 | 304 | 11.8 | |
| | *Missing*[d] | 2,816 | | 2,190 | | 626 | | |
| | Number of symptoms (IQR) | 1 (0, 2) | | 1 (0, 2) | | 1 (0, 2) | | |
| **Number of HES admissions (IQR) Range of admissions** | | 3 (1, 5) 1–1,413 | | 3 (2, 6) 1–951 | | 1 (1, 5) 1–1,413 | | |

[a]At time of first CRIS admission.

[b]Nearest to first general hospital admission.

[c]HoNOS domains closest to general hospital admission.

[d]HoNOS domain with most missing data.

CRIS, Clinical Record Interactive Search; HES, Hospital Episode Statistics; HoNOS, Health of the Nation Outcome Scale; IQR, interquartile range; SD, standard deviation.

disorders and 67.1% (66.2–67.9) for bipolar affective disorders. Category-specific admission-level sensitivity was 45.5% (45.0–46.1) for admissions of people with schizophrenia spectrum disorder (F20-29) and 39.6% (38.7–40.5) for bipolar affective disorders. Admission-level sensitivity according to primary diagnosis varied from 33.1% (31.3–34.9) for admissions related to pregnancy, childbirth, and the puerperium (ICD-10 codes O00-O99) to 79.7% (78.4, 81.0) for

**Table 2. Sensitivity of general hospital diagnoses of SMI 2006–17, at the level of each individual patient's whole-hospital records.**

| Psychiatric Diagnosis | HES Diagnosis | Number of True Positives | Sensitivity % (95% CI) |
|---|---|---|---|
| | Any F code | 10,574/13,786 | 76.7 (76.0–77.4) |
| F20-29 | % with any F | 7,796/10,061 | 77.5 (76.7–78.3) |
| | % with F20-29 | 5,672/10,061 | 56.4 (55.4–57.4) |
| F30-31 | % with any F | 2,778/3,725 | 74.6 (73.2–76.0) |
| | % with F30-31 | 1,852/3,725 | 49.7 (48.1–51.3) |

F20-29 Schizophrenia, schizotypal and delusional disorders (excluding acute and transient psychotic disorders; and induced delusional disorder); F30-31 Manic episodes and bipolar affective disorder.

CI, confidence interval; HES, Hospital Episode Statistics; SMI = severe mental illness.

poisoning-related admissions (T36-65) and 81.0% (79.0, 82.9) for endocrine, nutritional, and metabolic diseases (E00-E90) (see S1 Table for full results).

## Time-trend changes from 2006 to 2017

Sensitivity of recording for any psychiatric diagnosis increased from 47.8% (43.1–52.5) for emergency admissions during 2006 to 75.4% (68.3–81.4) for admissions during 2017 ($p_{trend} <$ 0.001 [$\chi 2 = 326$, 1 df]), although much of this change was observed between 2009 and 2012 (Fig 2; full data in S4 Table).

## Association of sociodemographic and clinical factors with psychiatric diagnosis being unrecorded

In unadjusted analyses, age, ethnicity, marital status, mental and physical symptoms, functional impairment, and number of hospital admissions were associated with diagnostic recording (Table 4). In mutually adjusted multivariable analysis, those from Black African/Caribbean backgrounds were more likely (odds ratio [OR] = 1.38 [95% CI 1.24, 1.55; $p < 0.001$]) to have no psychiatric diagnosis ever recorded compared with those from White ethnic backgrounds. Marital status was also associated with diagnostic accuracy; single people were less likely (OR = 0.78 [95% CI 0.63–0.92; $p < 0.001$]) to have no psychiatric disorder recorded compared with married individuals, as were divorced (OR = 0.76 [95% CI 0.63–0.92; $p = 0.004$]) or widowed people (OR = 0.77 [95% CI 0.60–1.00; $p = 0.046$]). More mental health symptoms were associated with greater diagnostic accuracy, with 2 (OR = 0.71 [95% CI 0.62–0.83; $p < 0.001$]) or 3 plus symptoms (OR = 0.61 [95% CI 0.52–0.73; $p < 0.001$]) being associated with lower

**Table 3. Sensitivity of general hospital diagnoses of SMI 2006–17, at the level of emergency hospital admissions only.**

| Psychiatric Diagnosis | HES Diagnosis | Number of True Positives | Sensitivity % (95% CI) |
|---|---|---|---|
| | Any F code | 32,033/45,706 | 70.1 (69.7–70.5) |
| F20-29 | % with any F | 24,143/33,941 | 71.1 (70.7–71.6) |
| | % with F20-29 | 15,453/33,941 | 45.5 (45.0–46.1) |
| F30-31 | % with any F | 7,890/11,765 | 67.1 (66.2–67.9) |
| | % with F30-31 | 4,660/11,765 | 39.6 (38.7–40.5) |

F20-29 Schizophrenia, schizotypal, and delusional disorders (excluding acute and transient psychotic disorders; and induced delusional disorder); F30-31 Manic episodes and bipolar affective disorder.

CI = confidence interval; HES = Hospital Episode Statistics; SMI = severe mental illness.

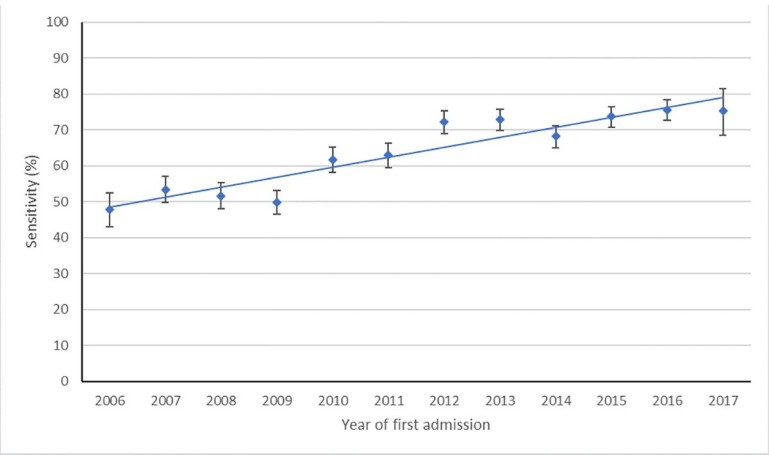

**Fig 2. Time-trend of sensitivity for schizophrenia spectrum disorders and bipolar affective disorders diagnosis in general hospitals.** Points represent the proportion of people's first emergency hospital admissions following severe mental illness diagnosis in which a mental illness is recorded. Error bars show 95% confidence interval, and linear trend line shows change over time.

**Table 4. Association of clinical and sociodemographic characteristics with psychiatric diagnosis of people with severe mental illness not being recorded in general hospital records: Univariate and multivariable logistic regression.**

| Sociodemographic and Clinical Characteristics | | | Unadjusted Analysis (*n* = 13,786) | | Mutually Adjusted Multivariable Analysis (*n* = 10,189) | |
|---|---|---|---|---|---|---|
| | | | Odds Ratio (95% CI) | *p* value | Odds Ratio (95% CI) | *p* value |
| Age (per 10-year increment) | | | 0.89 (0.87–0.91) | <0.001 | 0.98 (0.94–1.01) | 0.180 |
| Sex | Female (Ref) | | 1 | | 1 | |
| | Male | | 0.93 (0.86–1.01) | 0.072 | 0.90 (0.82–1.00) | 0.054 |
| Ethnicity | White (Ref) | | 1 | | 1 | |
| | Asian | | 1.35 (1.13–1.62) | 0.001 | 1.10 (0.87–1.38) | 0.424 |
| | Black African/Caribbean | | 1.51 (1.38–1.66) | <0.001 | 1.38 (1.24–1.55) | <0.001 |
| | Mixed | | 1.33 (1.03–1.72) | 0.027 | 1.26 (0.93–1.71) | 0.136 |
| | Other | | 1.64 (1.38–1.94) | <0.001 | 1.25 (1.00–1.55) | 0.047 |
| Marital Status | Married (Ref) | | 1 | | 1 | |
| | Single | | 0.78 (0.70–0.87) | <0.001 | 0.72 (0.62–0.82) | < 0.001 |
| | Divorced | | 0.71 (0.61–0.83) | <0.001 | 0.76 (0.63–0.92) | 0.004 |
| | Widowed | | 0.57 (0.46–0.70) | <0.001 | 0.77 (0.60–1.00) | 0.046 |
| Deprivation Score (per 10-unit increase) | | | 1.01 (0.98–1.05) | 0.467 | 1.01 (0.96–1.06) | 0.824 |
| Clinical symptoms and function (Health of the Nation Outcome Scale) domains | Mental Health Subscale | No Symptoms (Ref) | 1 | | 1 | |
| | | 1 Symptom | 0.83 (0.75–0.93) | 0.001 | 0.90 (0.79–1.01) | 0.081 |
| | | 2 Symptoms | 0.66 (0.58–0.75) | <0.001 | 0.71 (0.62–0.83) | <0.001 |
| | | 3+ Symptoms | 0.53 (0.46–0.61) | <0.001 | 0.61 (0.52–0.73) | <0.001 |
| | Problem with Physical Illness | | 0.55 (0.50–0.60) | <0.001 | 0.84 (0.74–0.95) | 0.004 |
| | Problem with Daily Living | | 0.54 (0.49–0.60) | <0.001 | 0.71 (0.62–0.80) | <0.001 |

Multivariable analysis adjusted for age, sex, ethnicity, marital status, deprivation score, clinical symptoms and function and log number of hospital admissions.

risk of no psychiatric diagnosis being recorded. Difficulties with activities of daily living (OR = 0.71; 95% CI 0.74–0.95; $p < 0.001$) and physical illness (OR = 0.84; 95% CI 0.74–0.95; $p = 0.004$) were also associated with lower risk of unrecorded diagnosis. Results were consistent in sensitivity analyses without inclusion of HoNOS symptoms (S5 Table) and with multiple imputation for missing covariates (S6 Table).

## Discussion

In this study, we examined accuracy of general hospital records for people with SMI admitted to general hospitals, finding that at least some form of psychiatric diagnosis was recorded in 70.1% of individual emergency hospital admissions and in 76.7% of patients' complete hospital discharge records. Our findings suggest that accuracy of SMI recording in general hospitals has improved over time with sensitivity for any psychiatric diagnosis in those experiencing emergency hospital admissions increasing from 47.8% in 2006 to 75.4% in 2017. Unrecorded psychiatric diagnosis was more likely in people with milder symptoms or higher ADL scores, married individuals, and ethnic minority groups.

Although there has been improvement in recording of SMI within the general hospitals investigated, our findings show there are nearly 25% of individuals with established SMI diagnosis who have never had a preexisting SMI recorded throughout their, on average, 3 subsequent general hospital admissions. Moreover, recording is lower for category-specific sensitivity. Thus, the proportion of individuals with schizophrenia spectrum disorders who received a diagnosis within that same ICD-10 F20-29 category was 56.4%, compared with 77.5% for recording of any psychiatric diagnosis, and for bipolar affective disorders, these figures were 49.7% and 74.6%, respectively. The lower sensitivity for more specific diagnoses may reflect uncertainty about diagnostic labels, including whether these are accurate and used or communicated appropriately [31]; there is the potential for disagreement between clinicians and for different diagnostic practices in different settings. The relatively low sensitivity at a category-level is potentially problematic, when considering the importance of specific diagnosis for treatment outcomes and continuity of care, such as the different symptom clusters, illness severity, and pharmacological treatment indicated for schizophrenia compared with, for example, an anxiety disorder [32].

We acknowledge, however, that it is not necessarily the role of a physician or surgeon in a physical healthcare setting to diagnose a specific psychiatric condition, and instead, recognition of the presence of a mental illness may suffice if it leads to a referral to more specialist mental healthcare service such as liaison psychiatry. Other holistic measures to promote the wellbeing of a general hospital inpatient with SMI may also be instituted after identification of the presence of a psychiatric disorder. These may include consultation with family to identify whether psychiatric symptoms are new or preexisting, discussion with community mental health teams, or provision of a side room offered so that sleep could be prioritized.

We are not aware of any previous studies that have examined sensitivity of psychiatric diagnosis recording in English general hospitals, so cannot compare our findings directly with others. Our findings are lower than the 88% and 96.3% coding accuracy reported for general practice EHR [10, 33] and deliberate self-poisoning admissions in UK hospital [34]. However, general practice records are likely to include lifelong diagnostic records, rather than be a snapshot of a clinical episode such as hospital admissions, so may be more likely to include a diagnosis received in another service sector. Self-poisoning is likely to have been the cause of presentation to hospital, and so recording of this is expected to be higher. Sensitivity in our study was higher than the figure of 52% reported for recognition of alcohol disorders by hospital staff [12].

Increased recording of diagnosis since the introduction of payment by results in 2006/2007 from 49.5% to 81.3% in 2017 is likely due to a range of different reasons from changes to policy

approaches such as the "Five-Year Forwards" plan for whole-person centered care [34], financial incentives [35], improvements in coding practices [35], and expansion of liaison psychiatric services [36], whereby psychiatric diagnoses are more readily available (whether assigned anew or confirmed from previous records) from mental health specialists.

The association between clinical symptoms and unrecorded psychiatric diagnosis showed a dose–response gradient. It is likely that increasing symptom severity reflects clinical complexity meaning that difficulties are more prominent, and diagnosis is therefore more likely to be recorded. Similarly, symptom severity might reflect more frequent or longer hospital admissions during which clinicians have more opportunity to investigate clinical records. These individuals might be better known to clinical staff as they have more regularly updated HES records. People with milder SMI are less likely to have acute psychiatric healthcare needs, so lower accuracy in milder cases may be of less concern.

Unrecorded diagnosis was more common in individuals from Black African/Caribbean or other ethnic minority backgrounds. There are several potential reasons for this, relating to patient or service-level factors [37]. Patient-level factors include language barriers, lack of information disclosure, or distrust of clinicians from different backgrounds, and service-level factors include clinician bias, stigma, or lack of cultural awareness [38]. Considering poor health outcomes for people from minority ethnic groups with SMI, including increased cardiovascular risk factors [39], reduced support post discharge [40], and increased rates of compulsory admissions [41], this is particularly concerning. Recording of SMI in general hospitals might be an early opportunity for support as this is a setting for key treatment decisions and an opportunity for enhanced continuity of care [9, 42]. We did not find association of diagnostic accuracy with Asian or mixed backgrounds, possibly because of the smaller sample size giving less statistical power or reflecting other factors such as socioeconomic status.

Counter to our expectation based on analyses of other mental disorders [43, 44], being married was associated with higher likelihood of diagnosis being unrecorded. This may be explained by marital status being a marker of SMI severity and chronicity, whereby symptoms are higher in those who are single as compared with those in relationships. However, adjustment for symptom severity did not attenuate this association. Alternatively, it may be that support from a partner makes mental health symptoms less apparent as clinicians might assume that someone is receiving support from home or feel less inclined to ask about symptoms of mental illness. Similarly, it might be that partners are less reluctant to disclose information in regard to SMI diagnosis because of increased stigma associated with such conditions.

Sensitivity appeared to vary according to the primary reason for admission so that admissions for poisoning (which may have been precipitated by mental health symptoms) or those for endocrine and metabolic conditions (which may result from psychotropic adverse effects) had higher diagnostic recording. The lower recording for pregnancy and childbirth-related is of concern and warrants further future investigation, though it may reflect coding practice differences.

Important strengths of this study include its large sample size, representativeness of people with SMI diagnosis, and data availability over a decade enabling analysis of changes over time. The main limitation is that EHRs are not primarily collected for research purposes, meaning that using SLaM records as the "reference standard" might be problematic because the amount of clinical contact before diagnosis, and the diagnostic process may have varied, so some of our cohort may have been misdiagnosed. However, SLaM provides specialist diagnostic services, and we grouped patients according to the most recent recorded diagnoses from either structured quantitative health outcomes or rich unstructured clinical records. This approach also allowed construction of a large cohort that was representative of people with clinically diagnosed SMI, which would not have been feasible had we interviewed all participants with a

standardized clinical assessment. Although there is wider debate about the validity of psychiatric diagnostic constructs [45], SMI categories are considered stable and persistent over time [46]. Furthermore, our primary analysis considered sensitivity at the level of all psychiatric diagnoses, for example, whether an individual with schizophrenia had any mental illness recorded, as we acknowledged the potential limitations of the reference standard.

Our cohort's derivation from secondary mental health services may have meant that participants had more pronounced symptoms and care-seeking behaviors, which would likely result in overestimation of sensitivity, but we included individuals with SMI in remission and included a range of clinical severity. Although HES records cover all English hospitals, most admissions will have been in South London hospitals. Diagnostic practice in rural settings where clinical populations and resources may differ. Finally, use of general hospital discharge diagnoses may miss nuances of diagnostic practice in general hospital in which the SMI may have been recognized during the admission but not recorded on final discharge, and our observational study design meant that we were unable to examine whether diagnostic recording affected clinical outcomes.

Future studies should examine whether diagnostic recording affects outcomes, such as length of stay or readmission rate, and has any negative effects such as stigma or diagnostic overshadowing of physical illness. This may require more detailed scrutiny of individuals' case records and could be analyzed as part of future evaluations of liaison/consultation psychiatry services. It is also important to consider other relevant metrics of diagnostic accuracy, such as positive predictive value or specificity, which would require reference-standard data representative of people without psychiatric illness. Future research should also aim to elucidate the mechanisms for association between ethnicity, marital status, and symptom severity and diagnostic recording in general hospitals and evaluate effective approaches to improving diagnostic practice.

Our findings have important clinical, research, and policy implications. Researchers wishing to use hospital EHRs such as HES to ascertain cases of schizophrenia-like illnesses and bipolar disorder should be aware that around one quarter of known cases are likely to be unrecorded and there is potential for systematic bias whereby ethnic minorities, married people, and people with milder symptoms will be missed. Although our findings suggest that sensitivity is improving over time, there were around 30% of admissions in which people with established SMI did not have any psychiatric diagnosis recorded, suggesting that more needs to be done by policymakers to bridge the gap for "whole-person centered" care. Hospital settings should endeavor to improve diagnostic recording, particularly for high-risk groups. This may include training staff in culturally sensitive diagnosis for ethnic minority populations, expansion of mental healthcare input in general hospitals and collaborative working with these liaison psychiatry services, and proactive contact with primary care and mental health services to elicit information about past psychiatric history. Better data sharing between physical and mental health services such as through harmonized clinical records could improve accuracy of mental illness in physical healthcare, and physical illness in mental healthcare services, to move towards truly integrated healthcare for people with mental illness.

## Supporting information

**S1 STROBE Checklist.** STROBE, Strengthening the Reporting of Observational Studies in Epidemiology.
(DOCX)

**S1 Text. Prospective analysis plan.**
(DOCX)

**S1 Table. Primary diagnosis (ICD-10 code) for emergency hospital admissions of people with severe mental illness.** ICD-10, International Statistical Classification of Diseases and Related Health Problems, Tenth Revision.
(DOCX)

**S2 Table. Clinical characteristics of people with severe mental illness admitted to a general hospital, according to whether mental illness was recorded in their hospital records.**
(DOCX)

**S3 Table. Disorder-specific sensitivity of general hospital emergency admission records for people with severe mental illness.**
(DOCX)

**S4 Table. Recording of mental illness in people with severe mental illness admitted to general hospitals: By year of first emergency hospital admission.**
(DOCX)

**S5 Table. Association of sociodemographic characteristics with recording of mental illness in general hospital records: Multivariable regression without Health of Nation Outcome Scale clinical characteristics.**
(DOCX)

**S6 Table. Association of clinical and sociodemographic characteristics with psychiatric diagnosis of people with severe mental illness not being recorded in general hospital records: Multivariable regression with multiple imputation for missing variables.**
(DOCX)

## Author Contributions

**Conceptualization:** Hassan Mansour, Christoph Mueller, Andrew Sommerlad.

**Data curation:** Hassan Mansour, Christoph Mueller, Hitesh Shetty.

**Formal analysis:** Hassan Mansour, Christoph Mueller.

**Funding acquisition:** Robert Stewart.

**Investigation:** Hassan Mansour, Christoph Mueller, Andrew Sommerlad.

**Methodology:** Hassan Mansour, Christoph Mueller, Katrina A. S. Davis, Alexandra Burton, David Osborn, Robert Stewart, Andrew Sommerlad.

**Project administration:** Hassan Mansour, Christoph Mueller, Hitesh Shetty, Robert Stewart.

**Supervision:** Christoph Mueller, Andrew Sommerlad.

**Visualization:** Hassan Mansour.

**Writing – original draft:** Hassan Mansour.

**Writing – review & editing:** Hassan Mansour, Christoph Mueller, Katrina A. S. Davis, Alexandra Burton, Matthew Hotopf, David Osborn, Robert Stewart, Andrew Sommerlad.

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
