## [Editor Report · Decision Letter 0]

11 Mar 2020

Dear Dr Mansour, 

Thank you for submitting your manuscript entitled "Severe mental illness diagnosis in English general hospitals 2006-2017: sensitivity, time-trends, and predictors of diagnostic accuracy" for consideration by PLOS Medicine.

Your manuscript has now been evaluated by the PLOS Medicine editorial staff and I am writing to let you know that we would like to send your submission out for external peer review.

Kind regards,

Helen Howard, for Clare Stone PhD 

Acting Editor-in-Chief

PLOS Medicine 

plosmedicine.org

---

## [Decision Letter · Decision Letter 1]

2 May 2020

Dear Dr. Mansour,

Thank you very much for submitting your manuscript "Severe mental illness diagnosis in English general hospitals 2006-2017: sensitivity, time-trends, and predictors of diagnostic accuracy" (PMEDICINE-D-20-00812R1) for consideration at PLOS Medicine. 

[LINK]

In light of these reviews, I am afraid that we will not be able to accept the manuscript for publication in the journal in its current form, but we would like to consider a revised version that addresses the reviewers' and editors' comments. Obviously we cannot make any decision about publication until we have seen the revised manuscript and your response, and we plan to seek re-review by one or more of the reviewers. 

We expect to receive your revised manuscript by May 25 2020 11:59PM. Please email us (plosmedicine@plos.org) if you have any questions or concerns.

We look forward to receiving your revised manuscript. 

Sincerely,

Emma Veitch, PhD

PLOS Medicine

On behalf of Clare Stone, PhD, Acting Chief Editor,

PLOS Medicine

plosmedicine.org

*We'd suggest revising the title according to PLOS Medicine's style, this should include the study design in the subtitle (ie, after a colon) - eg here ": registry linkage study" (or similar).

*Please restructure your abstract using the PLOS Medicine headings (Background, Methods and Findings, Conclusions). The writeup for each subsection should be complete sentences (rather than sentence fragments).

*At this stage, we ask that you include a short, non-technical Author Summary of your research to make findings accessible to a wide audience that includes both scientists and non-scientists. The Author Summary should immediately follow the Abstract in your revised manuscript. This text is subject to editorial change and should be distinct from the scientific abstract. Please see our author guidelines for more information: https://journals.plos.org/plosmedicine/s/revising-your-manuscript#loc-author-summary

*We'd suggest ensuring that the study is reported according to the STROBE guideline (for observational studies), and include the completed STROBE checklist as Supporting Information. 1 Please add the following statement, or similar, to the Methods: "This study is reported as per the Strengthening the Reporting of Observational Studies in Epidemiology (STROBE) guideline (SChecklist)." The STROBE guideline can be found here: http://www.equator-network.org/reporting-guidelines/strobe/. When completing the checklist, please use section and paragraph numbers, rather than page numbers.

*Did your study have a prospective protocol or analysis plan? Please state this (either way) early in the Methods section.

Comments from the reviewers:

Reviewer #1: Thanks for the opportunity to review your manuscript. My role is as a statistical reviewer, and so my comments and queries focus on the data and the analysis in this work (and the presentation of these). This manuscript aims to estimate the sensitivity of diagnosis for serious mental illness during general hospital admissions (i.e. not to specialist mental health units), and whether this varies according to patient characteristics. 

I think this is a well-written and presented manuscript. The sensitivity analyses are appropriate and anticipated my initial thoughts about the missing data issues we usually encounter from routinely-collected data.

The analyses looking at how patient-level factors are associated with missed diagnosis are estimated using logistic regression. The description of the odds ratios, i.e. 'those from Black African/Caribbean backgrounds were 1.38 … times more likely' is incorrect, as an odds ratio will be a biased estimator of a rate ratio where the outcome (missed recording of the mental health diagnosis) is very common (e.g. ~50% overall). Either the analyses should be updated to estimate a rate ratio (i.e. log-binomial model or similar) or the wording should be adjusted when describing the odds ratios throughout the manuscript.

I found it slightly confusing to go from the outcome being sensitivity in the first part of the manuscript but the regression analyses then invert the outcome to be missed diagnoses. 

Specific comments:

P4, Paragraph 1. There are details of the source database and the linkage method used, is there any information available on the quality assurance procedures and the estimate false positive rate of record matching?

P6, Paragraph 2. Is the 'chi square test for trend' the Cochrane-Armitage test?

P6, Paragraph 3. What was the fraction of missing information found after imputation, and is the number of imputed datasets appropriate for the observed FMI?

Which version of Stata was used?

What type of model was used for each of the variables in the MI with chained equations? i.e. for categorical variables was this logistic or discriminant analysis? 

P7, Paragraph 4. What was the association between number of hospital admissions, and how would this be interpreted from the regression model/Table 4 given it's log-transformed? (also p25 appendix 5)

P7, Paragraph 5. There is an overall increase in trend but Figure 2 reveals that most of the change was from 2009 to 2012, rather than a linear increase over time.

P11, Paragraph 3. Is there a way with the datasets to zoom in specific hospital admissions where an accurate mental health diagnosis would equivocally be needed? This is more of a curious question than something that needs addressing the manuscript.

P22, Appendix 2. One of the few drawbacks to a large dataset like this is that Chi-square tests effectively just signal a large sample size - I would suggest including a measure of effect size (RR/OR) with corresponding CIs as well as a p-value and summary data.

P22, Appendix 4. Is the CI for the sensitivity (here and elsewhere in the MS) from a normal-approximation method or something else?

Reviewer #2: Thanks for the opportunity to peer review this well-conducted study.

It addresses an important topic - the disparities in mortality and morbidity, largely due to preventable medical conditions, that are experienced by people with severe mental illnesses, and the role of health service integration and diagnostic practice as mechanisms for this health inequality. 

The study uses electronic medical records to link specialist mental health service users' clinical and sociodemographic profiles with their episodes of care in emergency medical facilities, and specifically focuses on the accuracy (sensitivity) of psychiatric diagnoses that are recorded in emergency medical care. 

Overall, this study makes good use of available data to provide examination of a key aspect of care: the methods, measures, and approach to analysis are clearly reported, the ability to adjust for the effects of a range of sociodemographic factors is particularly informative (and the findings and implications for minority ethnic groups with SMI are rightly highlighted), as is the exploration of time trends in diagnostic accuracy over the ten year study period. 

My main concern relates to the extent to which this measurable aspect of care - explicit psychiatric diagnosis recorded during the general medical /emergency episode - may be associated with differential clinical outcome. It clearly is an accessible indicator of integrated care. But a little more consideration of the impacts of this being noted or omitted would be helpful. The authors stress the importance of 'specific [psychiatric] diagnosis for treatment outcomes and continuity of care'.. and note that this may facilitate '..a referral to more specialist mental healthcare service such as liaison psychiatry'.. and that it will enable '..continuity of previous care such as medication, and tailoring of inpatient and discharge plans to account for comorbidity'. 

These are important principles of care, but, could some more practical/ concrete inference or example be made? Could it be useful to consider the types of presenting condition in the emergency setting to clarify relationships between diagnosis and care? e.g. Appendix 1 - indicates common primary diagnoses to be injury and poisoning, - where mental ill-health may be directly related to event or risk.. or endocrine/metabolic disease where anti-psychotic medication may be especially relevant.. or maybe a broader statement that for many of the presenting problems, psychotropic medications may be directly relevant to management decisions such as prescribing. 

I have some minor suggestions for the introduction section.

The initial quantification of SMI prevalence ('..affect[s] around 3% of the population..') uses lifetime prevalence derived from a single, albeit high quality, general population study in Finland; it could be useful to clarify the statement - at least by noting this rather high estimate is for lifetime prevalence, and the authors might consider alternatively noting the point or period estimates available from systematic review/s.

The authors note limited studies concerning mental disorder recognition in medical settings, but may wish to modify their statement about the paucity of research in this area, noting that Mitchell et al (2012) have a conducted systematic review of alcohol misuse identification (2012), and they might also allude to the larger body of work (some of which they later cite) on psychosis recognition in primary care.

Reviewer #3: This analysis aimed to examine the recording of SMI in secondary physical healthcare settings (general hospitals) compared to that recorded in secondary mental healthcare in (1) sensitivity, (2) time trends, (3) factors associated with accurate coding. The paper is nicely structured and written. 

In the rationale, the authors describe that understanding coding of psychiatric disorders in general hospitals will facilitate our understanding of the healthcare received for physical needs in people with SMI, i.e. from HES can we identify people with SMI who are in hospital for any reason, and how accurate the diagnostic labels are. I agree this is important, as physical health problems appear to be a main driver of the mortality and morbidity gap. 

Unfortunately the analysis carried out is sub-optimal to address this aim.

The main problem is that the primary diagnosis in HES relates to the reason for admission. "The primary diagnosis is the main reason the patient is receiving care in hospital, while the secondary diagnoses are relevant co-morbidities and external causes if these have been identified." - this is taken from a HES metadata guide (available here https://digital.nhs.uk/data-and-information/publications/statistical/hospital-admitted-patient-care-activity/2018-19)

Accordingly, someone with SMI being admitted for a respiratory condition, for example, would have a primary code from ICD-10 chapter J, and may have a secondary code about their psychiatric diagnosis from chapter F (if it was recorded!). By focusing only on the primary code, you have primarily selected on psychiatry related admissions, and not on admissions with a primary indication for physical health problems. The HONOS data by recording status in Table 1 and Appendix 6 appear to indicate this. Because your rationale is based on physical health problems, I do not think this was your intention. If it was, then please reframe the paper accordingly (i.e. you wanted to look at psychiatric codes for psychiatric problems admitted through emergency admissions in general hospitals).

My main recommendation is to re-analyse the data using the secondary codes in HES, and then consider the effect of the primary versus secondary codes, perhaps through sensitivity analyses. You could further investigate the issues of coding with reason for admission with the use of consultant speciality codes. 

Although this means reconstructing and re-analysing the whole dataset, the value of your findings the research community is such that this is well worth doing. If this is not done, then I think there are only very limited information to be gained as your HES analytic sample is overly selected.

The below relates to a revised analysis. 

Query the terminology of 'gold-standard' used to describe the diagnosis applied in the SLaM data, perhaps 'reference standard' is more appropriate? What is the 'right' label and how it is arrived at (and who decides) in this context is appropriate to be elaborated on, however briefly, and there is a multitude of literature on this. 

Could the objectives be made more clear in that they relate to the general hospitals, e.g. … association of sociodemographic and clinical factors with [more] accurate psychiatric recording [in general hospital records]

Methods

How much influence did the hierarchical classification have? How did you operationalise this, for example if someone has 2 diagnoses in SLaM, but you found a diagnosis of bipolar disorder in HES and they were assigned as having schizophrenia in your algorithm, was this counted as missed code? You find that sensitivity for F31 is lower than for F20, how much of this is due to the method? I may be missing something but I don't see why this classification is necessary if some people have more than one SMI, please either justify, or do not apply the hierarchy. It would be helpful if you provided the number with more than one diagnosis code in SLaM.

Further elaboration on HES, for example its function and purpose, who enters the data, how the diagnoses are arrived at, the purpose and structure of diagnostic data contained within it (e.g. primary versus secondary), and any changes to the recording of these data over the studies time period are needed. 

You could move the rationale behind focusing on emergency admissions from the sensitivity analysis to the main methods section on HES. Please reference the evidence for rationale. 

How did the covariates of age, sex, ethnicity and marital status get into the SLaM records? E.g. via an admissions interview, or self-reported questionnaire, or from relatives, or a mixture? Are there any indications of its accuracy?

Please note which address was used to generate IMD, e.g. first, last contact in the dataset? It isn't usual to analyse an IMD mean score; usually IMD is categorised into quintiles or deciles (quartiles for very small datasets).

Please consider how the use of multiple admissions data can be analysed (see point in Results and Discussion below)

Results and discussion

In figure 1, please add the final number of people with SMI with a discharged admission (i.e. analysed number). It is noted that 13,786 people were admitted, but please clarify in the figure that when the 25,941 discharged admissions were discarded that none of your sample were lost?

Page 7 under Sensitivity of general hospital SMI diagnosis: "We found that 10,574 of 13,786 people with SMI *were diagnosed with any psychiatric illness* during their…" I don't think this is what you meant, but the terminology is also used in the discussion, and in the Interpretation section of the Abstract. The distinction of being 'diagnosed' with SMI in hospital versus 'a pre-existing SMI being noted' could be more clear and consistent. They may both be true, in which case this needs commenting on. As such, it becomes important to understand and comment further on *how* existing diagnoses make it into HES related to patient / informant disclosure, or linkage to other systems? Is there any literature to bring in here? How these diagnoses are recorded is central to your analysis, so consider introducing some of what you write across page 10/11 into the introduction. 

Related to this is the important point you make about a previous admission being a source of information for a current admission. Did you not consider adding this to your analysis plan? Even though you don't have true 'incident' admission, you might be able to pick this up as a signal. If your aim is to help researchers understand SMI recording in HES better, then pointing them to this potential source of under-reporting in more distant versus recent more admissions might be important.

[LINK]

---

## [Decision Letter · Decision Letter 2]

9 Jul 2020

Dear Dr. Mansour,

Thank you very much for re-submitting your manuscript "Accuracy of severe mental illness diagnosis in English general hospitals 2006-2017: a registry linkage study" (PMEDICINE-D-20-00812R2) for review by PLOS Medicine.

I have discussed the paper with my colleagues and it was also seen again by two reviewers. I am pleased to say that provided the remaining editorial and production issues are dealt with we are planning to accept the paper for publication in the journal.

[LINK]

If you have any questions in the meantime, please contact me (cmoyer@plos.org) or the journal staff at plosmedicine@plos.org. 

We look forward to receiving the revised manuscript by Jul 16 2020 11:59PM. 

Sincerely,

Caitlin Moyer, Ph.D.

Associate Editor 

PLOS Medicine

plosmedicine.org

Requests from Editors:

1.Data Availability statement: You have noted that “Due to the nature of the data, open access is not possible. However, data is available subject to the standard conditions of the CRIS at Maudsley database. Please contact the CRIS administrator for details.” Please provide contact information (email address and/or web link) for the CRIS administrator.

2.Abstract: Methods and Findings: As noted in response to reviewer 3, consider replacing the term “gold standard” with “reference standard”

3.Abstract: Conclusions: We suggest revising the first sentence to “Our findings suggest that there have been improvements in recording of SMI diagnoses, but concerning under-recording,

especially in minority ethnic groups, persists.” or similar.

4. Author Summary: Why was this study done?: Please revise to:

--People with severe mental illness (SMI) have increased mortality and morbidity, largely due to preventable medical conditions, and these disparities have the potential to be ameliorated through better healthcare integration.

5. Author Summary: What do these findings mean?: Please revise to:

--A limitation of our study is that our use of electronic health records for the reference-standard means that some people with SMI may have been misclassified.

6. Introduction: Final paragraph, bottom of page 4: Please describe the specific objectives of your study in paragraph form, rather than presenting as a numbered list.

7.Methods: Please specify the nature of participant consent, including whether informed consent was written or oral.

8.Methods: Page 8: Please remove the word “the” in the following sentence: “The We then conducted logistic regression on each imputed dataset before combining coefficients using Rubin’s rules.”

9.Results: top of page 10: We suggest revising this sentence for clarify, to: “Marital status was also associated with diagnostic accuracy; single people were less likely (OR = 0.78 (95% CI 0.63, 0.92; p < 0.001)) to have no psychiatric disorder recorded compared to married individuals, as were divorced or widowed people.” Please include the OR, 95% CIs and p values for diagnostic accuracy for individuals who are divorced or widowed.

10.Discussion: first paragraph: We suggest revising the following sentence to “Our findings suggest that accuracy of SMI recording in general hospitals has improved over time with sensitivity for any psychiatric diagnosis in those experiencing emergency hospital admissions increasing from 47.8% in 2006 to 75.4% in 2017.” or similar

11.Discussion: first paragraph: We suggest clarifying this sentence by revising to: “Unrecorded psychiatric diagnosis was more likely in people with milder symptoms or higher ADL scores, married individuals, and ethnic minority groups.” or similar

12.Discussion: Please remove sub headings from the Discussion section (e.g.“strengths and limitations”)

13.Discussion: Page 13: Please spell out the abbreviation for PPV.

14.Discussion: Top of page 14: We suggest revising to: “Whilst our findings suggest that sensitivity is improving over time, there were around 30% of admissions where people with established SMI did not have any psychiatric diagnosis recorded, suggesting that more needs to be done by policymakers to bridge the gap for ‘whole-person centered’ care”

15. Discussion: Page 14: We suggest revising to: “Better data sharing between physical and mental health services such as through harmonised clinical records could improve accuracy of mental illness in physical healthcare, and physical illness in mental healthcare services, to move towards truly integrated healthcare for people with mental illness.”

16. Declaration of interests, Ethical Approval, and Data Sharing: Please remove these sections from the manuscript and make sure that the relevant information is included in the “Competing Interests” , Ethics Statement, and Data Availability sections of the manuscript submission form.

17. Reference list: Please use the "Vancouver" style for reference formatting, and see our website for other reference guidelines https://journals.plos.org/plosmedicine/s/submission-guidelines#loc-references

18. Figure 1: Please include a descriptive legend. Please spell out abbreviations for SMI, CRIS, ICD-10, HES. Please include the different F codes in the legend, or indicate where they can be found.

19.Figure 2: Please include a legend describing the figure, including an explanation of the trend line and what is represented by the error bars.

20. Table 1: Please include 95% CIs in addition to p values.

21. Table 2 and Table 3: Please include a descriptive legend. Please define abbreviations for SMI, HES, CI. Please describe the range of F code diagnosis for each category.

22. Table 4: Please indicate in the legend, the factors that are adjusted for in the adjusted analyses.

23. S2 Table: Please update the “significance test” column with what is being described (please include the test statistic, 95% CIs and p values). Please indicate if the values for other columns indicate n and percent.

24. S5 Table and S6 Table: Please also include the unadjusted results. In the legend, please indicate which factors are adjusted for in the analyses.

25. In-text citations throughout the manuscript: Instead of superscript notation, please place in-text citations in brackets before punctuation, like this [1].

Comments from Reviewers:

Reviewer #1: Thanks for the revised manuscript and detailed responses to my comments. I am happy with the revisions made and from the statistical side I don't think any further changes need to be made.

94% succesfully matching overall is reasonable for admission data with deterministic matching and I'm comfortable with the data quality in this context. The presenteation of the logged admission data is more easily understood now. The 'missed diagnosis' part of the analysis is clearer now and I agree that it does not need to be changed further. Supp table 1 is interesting it's reassuring that MH diagnosis are accurately being captured for in mental and behavioural disorders admissions. I don't think this needs further comment but hopefully this can be a useful resource for other researchers interested in metnal health in other conditions (the low rate in preganancy/childbirth is worrying!). 

Reviewer 3 has made some valuable comments on this manuscript and it sounds like they have familiarity with this particular dataset. Your responses look reasonable to me and I'm interested in seeing R3's take on these.

Reviewer #3: I am satisfied that the authors have addressed my concerns. This paper is a useful and careful contribution to the literature.

[LINK]

---

## [Editor Report · Decision Letter 3]

21 Jul 2020

Dear Mr Mansour, 

On behalf of my colleagues and the academic editor, Dr. Stephanie Prady, I am delighted to inform you that your manuscript entitled "Accuracy of severe mental illness diagnosis in English general hospitals 2006-2017: a registry linkage study" (PMEDICINE-D-20-00812R3) has been accepted for publication in PLOS Medicine. 

PRODUCTION PROCESS

PRESS

PROFILE INFORMATION

Thank you again for submitting the manuscript to PLOS Medicine. We look forward to publishing it. 

Best wishes, 

Caitlin Moyer, Ph.D.

Associate Editor 

PLOS Medicine

plosmedicine.org